# Comparison of Root Transcriptomes against Clubroot Disease Pathogens in a Resistant Chinese Cabbage Cultivar (*Brassica rapa* cv. ‘Akimeki’)

**DOI:** 10.3390/plants13152167

**Published:** 2024-08-05

**Authors:** Eun-Seok Oh, Hyeonseon Park, Kwanuk Lee, Donghwan Shim, Man-Ho Oh

**Affiliations:** 1Department of Biological Sciences, College of Biological Sciences and Biotechnology, Chungnam National University, Daejeon 34134, Republic of Korea; eunseokoh0318@gmail.com (E.-S.O.); bhs2880@gmail.com (H.P.); 2Department of Biology, College of Natural Sciences, Jeju National University, Jeju 63243, Republic of Korea; kulee@jejunu.ac.kr

**Keywords:** clubroot disease, *Plasmodiophora brassicae*, Akimeki, transcriptomics, disease resistance, *Brassica rapa*

## Abstract

Clubroot, caused by *Plasmodiophora brassicae*, is one of the diseases that causes major economic losses in cruciferous crops worldwide. Although prevention strategies, including soil pH adjustment and crop rotation, have been used, the disease’s long persistence and devastating impact continuously remain in the soil. CR varieties were developed for clubroot-resistant (CR) Chinese cabbage, and ‘Akimeki’ is one of the clubroot disease-resistant cultivars. However, recent studies have reported susceptibility to several Korean pathotypes in Akimeki and the destruction of the resistance to *P. brassicae* in many Brassica species against CR varieties, requiring the understanding of more fine-tuned plant signaling by fungal pathogens. In this study, we focused on the early molecular responses of Akimeki during infection with two *P. brassicae* strains, Seosan (SS) and Hoengseong2 (HS2), using RNA sequencing (RNA-seq). Among a total of 2358 DEGs, 2037 DEGs were differentially expressed following SS and HS2 infection. Gene ontology (GO) showed that 1524 and 513 genes were up-regulated following SS and HS2 inoculations, respectively. Notably, the genes of defense response and jasmonic acid regulations were enriched in the SS inoculation condition, and the genes of water transport and light intensity response were enriched in the HS2 inoculation condition. Moreover, KEGG pathways revealed that the gene expression set were related to pattern-triggered immunity (PTI) and effector-triggered immunity (ETI) mechanisms. The results will provide valuable information for developing CR cultivars in Brassica plants.

## 1. Introduction

Clubroot, caused by the obligate biotrophic *Plasmodiophora brassicae*, is one of the most severe soil-borne diseases in cruciferous crops, including oilseed rape, radish, Chinese cabbage, mustard, and broccoli [1,2]. The life cycle shows that *P. brassicae* first penetrates root hairs and then the root cortex of the infected plant, affecting cell elongation as well as cell division, resulting in root swelling and gall formation [3,4]. Because spores of *P. brassicae* can maintain the infection in the soil for a period ranging from 17 to 20 years, clubroot disease has caused significant economic losses to cruciferous production for many years by blocking the water and nutrient uptake [1,5] and has led to severe damage to Brassica vegetable and oilseed crops worldwide each year [5]. It has been reported that each crop loses approximately 10 to 15% of its production. In particular, yield is reduced by 60 to 90% in fields infected with clubroot in susceptible rapeseed/rapeseed seeds [6].

To date, diverse management measures have been developed to control clubroot disease in infected soils. It is required to select a disease-resistant variety and to prepare the soil through various preventive measures. The preventive strategies, including pH adjustments of the soil, crop rotation, and application of chemicals, have been proposed and applied [7,8,9,10,11]. Although these methods reduce and prevent clubroot infection, the effects are still limited. Currently, an effective approach that reduces the use of chemical fungicides while reducing crop losses through the development of clubroot-resistant (CR) varieties is recognized as the most desirable method [12,13]. Despite many efforts to develop CR varieties, the destruction of resistance to *P. brassicae* has been recently reported in many species of Brassica, which may be due to mutations in the pathogen [14,15]. 

In higher plants, pathogen resistance mainly consists of two mechanisms, specific resistance to specific pathogens and basal resistance, both types of which are important in protecting plants from pathogens [16]. The molecular biology aspects of specific pathogen-specific resistance have been extensively studied and determined in promoting the expression of resistance genes by recognizing specific effectors of pathogens [17]. In addition, many previous studies and traditional breeding methods have identified several varieties with resistance to clubroot disease and, conversely, pathogens that cause clubroot disease have been found in resistant varieties [18]. Pathogen-associated molecular patterns, pattern-triggered immunity (PTI) and effector-triggered immunity (ETI), are considered as separate cycles of plant immunity in the zig-zag model [19]. In the PTI phase, exogenous triggers, PAMPs, can be components of the pathogen’s cell wall and membrane as well as bacterial flagellin (flg22) and EF-Tu (elf18), which are recognized by pattern recognition receptors (PRRs) [20]. These triggers can be released under the action of the host’s enzymes or secreted by pathogens during host–pathogen interactions [21]. To subvert PTI and to facilitate invasion, pathogens inject virulence effectors secreted via bacterial type 3 secretion machinery into plant cell or host cytoplasm, resulting from effector-triggered susceptibility (ETS) [19,22]. During evolution, plants have recognized the effectors via internal receptor proteins, such as nucleotide-binding and leucine-rich repeat (NB-LRR) receptors as R proteins, leading to ETI associated with programmed cell death (PCD), also known as the hypersensitive response (HR), and systemic-acquired resistance (SAR) [22]. This suggests that it is important to understand the defense mechanism involved in CR during either or both PTI and ETI. 

Recent studies have demonstrated that the genes related to cell wall biosynthesis as well as Ca^2+^, reactive oxygen species (ROS), salicylic acid (SA), ethylene (ET), jasmonic acid (JA), and the brassinosteroid (BR) signaling pathway are dramatically up-regulated in CR Brassica lines [23,24,25]. Interestingly, the study of infection signaling by fungal pathogens revealed that the chitin-induced mechanism starts with the chitin receptor CERK1. The chitin-signaling pathway induces the expression of early chitin-responsive genes, such as protein kinases (PKs), WRKYs, ethylene response factors (ERFs), and zinc finger (ZFs) via the mitogen-activated protein kinase (MAPK), and regulates the expression of defense genes, such as pathogenesis-related proteins (PRs), receptor-like kinases (RLKs), and nucleotide-binding site and leucine-rich repeat (NB-LRRs) [26]. Therefore, it is crucial to understand plant–pathogen interactions, plant immune systems, and different signaling pathways in clubroot, which can provide diversity in CR development.

‘Akimeki’ is a clubroot disease-resistant Chinese cabbage developed by Japanese researchers to be resistant to several *P. brassicae* strains and possesses Crr1, Crr2, and CRb resistance loci [27]. Although Korean researchers have tried to develop a CR genotype with resistance to the Korean pathogen strains, the variety is susceptible to several Korean pathogens [7]. In this study, we analyzed the global gene expression profiling of the early plant immune response of the CR cultivar ‘Akimeki’ to two *P. brassicae* strains isolated from different regions, Heongseong2 (HS2) and Seosan (SS), in South Korea to understand the immune response of CR to pathogenic and non-pathogenic strains through transcriptomic analysis. Remarkably, the findings reveal that the difference gene set is mainly involved in PTI and ETI mechanisms. Our results can confer a direction for the development of resistant cultivars by analyzing gene expression patterns during successful pathogen defense and susceptibility events and provide insights into the development of resistant cultivars.

## 2. Results

### 2.1. P. brassicae Infection and RNA Sequencing

‘Akimeki’, developed as a resistant variety, is known to be resistant to *P. brassicae* strains. Among the *P. brassicae* strains, no gall was observed in the roots of the Chinese cabbage after 5 weeks of inoculation with the Hoengseong2 isolate (HS2), but it was not resistant to the Seosan isolate (SS) and gall formation occurred (Figure 1). 

We treated two *P. brassicae* strains (SS and HS2) and profiled the early gene response of clubroot-resistant Akimeki 72 h after inoculation with the genome-wide transcriptome analysis. The RNA-seq data were generated using the Illumina paired-end method, with a paired reads rate exceeding 95%. The overall alignment rate was approximately 85%, verifying that the data are of high quality for subsequent analyses (Appendix A). The RNA-seq data were obtained from the roots of plants treated with Mock, SS, and HS2 inoculations, each with three biological replicates. The data quality was evaluated through PCA analysis and correlation analysis (Appendix A). For each comparison, the TMM-normalized TPM value with a fold-change value greater than 4 and an FDR P-value lower than 0.001 was considered a DEG. There were 2358 total DEGs, and hierarchical analysis between the conditions showed that there was a difference in the overall gene expression following the HS2 inoculation condition compared to the Mock and SS inoculation conditions (Figure 2a). Among the total DEGs, 2037 DEGs were differentially expressed following HS2 and SS inoculation, accounting for 86% of the total DEGs (Figure 2b).

### 2.2. Functional Enrichment Analyses of Differentially Expressed Genes

We performed a GO analysis on 1524 and 513 genes that were up-regulated in SS and HS2 inoculations, respectively, out of 2037 DEGs. From the 2037 DEGs identified through SS and HS2 inoculation, 1962 homologous genes in A. thaliana were used for GO enrichment analysis (Appendix A). Following the SS inoculation condition, Akimeki that subsequently formed root nodules were statistically significantly enriched in GO terms, corresponding to the defense response (Figure 3a). We identified 956, 61, and 38 genes corresponding to the GO terms defense response (GO:0006952), regulation of jasmonic acid-mediated signaling pathway (GO:2000022), and water transport (GO:0006833), respectively, and determined how they are enriched differentially in the whole-transcriptome data. Notably, gene set enrichment analysis (GSEA) further confirmed that defense and jasmonic acid regulation-related GO terms were up-regulated and enriched following the SS inoculation condition compared to the Mock condition and HS2 inoculation condition (FDR q-value = 0.0) (Figure 3c), whereas GO terms involved in water transport were up-regulated and enriched following the HS2 inoculation condition compared to the Mock and SS inoculation conditions (FDR q-value = 0.0) (Figure 3d), indicating that the defense response and JA signaling pathway genes are specifically up-regulated following SS inoculation and water transport in HS2 inoculation, respectively. The Mock treatment and HS2 inoculation specifically resulted in 231 DEGs, which corresponded to 215 *B. rapa* genes that were analyzed against 207 *A. thaliana* genes using BLAST for GO and KEGG enrichment analysis. In the Biological Process category, the term regulation of DNA-templated transcription (GO:0006355) was the most significantly enriched, and the term toxin catabolic process (GO:0009407) was also identified as enriched (Appendix A).

To investigate the differences following the HS2 and SS inoculation conditions in defense responses, we confirmed the expression of genes corresponding to the plant–pathogen interaction pathway of *B. rapa* (ath04626) among KEGG pathways by a heatmap (Figure 4). A total of 514 *B. rapa* genes were assigned to the pathway, and the expression of 160 defense-related genes among them is visualized in Figure 4. Among these, 40 were included from the 2037 DEGs identified through HS2 and SS inoculations. The expression of most genes within the overall pathway was identified to be up-regulated following the SS inoculation and down-regulated following HS2 inoculation. A few genes involved in pattern-triggered immunity mechanisms, such as cyclic nucleotide-gated ion channel, and calcium signaling-related genes were up-regulated following HS2 inoculation (Figure 4a,b). Some of the PTI1-like tyrosine protein kinases in the pathway corresponding to effector-triggered immunity mechanisms were up-regulated in the HS2 inoculation. In addition, some WRKY transcription factors were up-regulated (Figure 4c).

Because the GO analysis was enriched in the SA- and JA-mediated signaling pathways following SS inoculation compared to HS2 inoculation (Figure 3), and phytohormones play a crucial role in the plant’s immune response [25], we next examined the expression patterns of key gene families within the SA and JA hormone pathways to elucidate the reasons behind the susceptible and resistant responses induced by the two *P. brassicae* isolates (SS and HS2) (Figure 5). In the resistant phenotype induced by the HS2 inoculation, the key genes of JA biosynthesis, LOX and AOS, were down-regulated, whereas COI1 (XM_009135144.3, XM_009135144.3, and XM_009145020), a major JA coreceptor of the JA pathway, was mostly up-regulated (Figure 5a). In addition, the PDF1.2 gene (XM_009107990.3), which is a marker for the ERF-branch of the JA defense network, was up-regulated. In the key gene families of the SA hormone (Figure 5b), which crosstalks with JA, the HS2 inoculation resulted in the up-regulation of ICS 1 (Isochorismate synthase 1) (XM_009106500.3), which is necessary for the metabolically active SA biosynthesis induced by pathogens, and NPR1 (XM_009110938.3), which bioactive salicylic acid can bind to its receptors to activate. Moreover, the PR2 gene, which produces the final product of the SA defense network, was down-regulated in the HS2 inoculation. Conversely, the NPR3 genes, which SA binds to in the unstressed state, were up-regulated in the SS inoculation. The WRKY70 gene (NM_001301918.1, XM_009105752.3, and XM_033290656.1), which activates SA-associated defense signaling, was down-regulated in the SS inoculation. Taken together, the results suggest that the crosstalk between SA and JA hormone signaling pathways plays a crucial role in defense responses to *P. brassicae* isolates.

### 2.3. Quantitative RT-PCR Validation

To validate the RNA-seq data, we conducted qRT-PCR analysis with 35 genes (Figure 6). PTI-related genes, such as PTI-like protein kinase 3, PAD4 (partial and indirect), BAK1, BIK1, and ERF020, were differentially increased in expression in the SS inoculation, and genes such as AUXIN UPREGULATED RNA (SAUR10), TGA3, and GALT4 were differentially decreased in the SS inoculation. The genes of GTPase-activating protein 5 and CYP93E6 were up-regulated in the HS2 inoculation. We calculated the correlation using the ddCt method (RNA-seq data of the control condition and BrACT1, the control gene). The expression of these genes in qRT-PCR and RNA-seq analysis was highly correlated, and it indicated the reliability of the RNA-seq results.

## 3. Discussion

Clubroot disease has had a detrimental effect on the productivity of cruciferous crops around the world, and various control methods have been tried to address the problem. The approach of breeding disease-resistant varieties has been developed by many researchers. ‘Akimeki’ was developed as a disease-resistant variety against *P. brassicae* in Japan [7,27], but the variety is susceptible to several Korean pathogens [7]. Previous studies have reported that *P. brassicae* infection occurs 12 to 24 h after inoculation with *B. rapa*, with primary infection occurring at 72 h and secondary infection occurring after 72 h [28,29]. In this paper, we performed a transcriptome analysis to gain a comprehensive understanding of how Akimeki responds to the susceptible SS and resistant HS2 of *P. brassicae* during the phase between primary and secondary infection.

A study has reported that Akimeki including Crr1, Crr2, and CRb resistance loci is resistant to several *P. brassicae* strains [27]. On the contrary, our current results show that Akimeki is not resistant to the Korean SS isolate (Figure 1), suggesting that the resistance mechanisms developed in the previous study did not work against the SS strain. The discrepancy may occur because the SS isolate was not considered during the development of the Akimeki cultivar or because the strain evolved in nature to evade the host’s defenses.

Jia et al. (2017) has reported that approximately 345 DEGs were up-regulated genes in susceptible plants, but the DEGs were down-regulated in resistant plant after *P. brassicae* inoculation [30]. Similarly, we showed that, of all the DEGs with a TMM-normalized TPM value with a fold-change of more than 4 and an FDR *P*-value of less than 0.001, about 71% are down-regulated genes following the HS2 inoculation, and about 70% of them are up-regulated genes following the SS inoculation (Figure 2). The result suggests that Akimeki, following the SS inoculation, is relatively differentially regulated and is associated with clubroot formation in the susceptible plants compared to the Mock and HS2 inoculation conditions.

Salicylic acid (SA), jasmonic acid (JA), and ethylene (ET) are key hormones in plants’ immune mechanisms [31,32,33]. SA positively modulates the plant defense against biotrophic pathogen infection, whereas JA and ET are synergistically involved in the plant defense against necrotrophic pathogen infection. Previous studies reported that the SA level in the resistant Alister root was more pronounced or was not different to that of susceptible Hornet in *Brassicas*, whereas the JA level was higher in the sensitive lines than that in the resistant lines during either the early response or the initiation of gall formation against *P. brassicae* [34]. On the other hand, a recent transcriptome and metabolome study on the root of *Brassica rapa*. determined that SA accumulation in the susceptible plants was higher than that in the resistant plants [35]. In this study, we performed the functional analysis on 2037 DEGs following SS and HS2 inoculation out of total DEGs. The results showing the gene ontology (GO) related to the biological process are mainly enriched with defense response-related terms and defense-related hormone terms (salicylic acid, jasmonic acid, and ethylene) following the SS inoculation (Figure 3). Moreover, SA and JA hormones were antagonistically regulated via the involvement of the SA- and JA-mediated signaling pathways and hormone biosynthesis, as well as the major regulators, such as NPRs, WRKYs, and MYCs [36,37,38,39,40,41,42,43,44]. In accordance with the previous studies, our current result shows that key genes involved in JA biosynthesis (LOX1, LOX2s, and AOS) were down-regulated by the HS2 inoculation, which does not induce clubroot, whereas some regulators (COI1_1, COI1_4, and COI1_5) and the final response gene (PDF1.2b_1) were up-regulated. Conversely, a key gene (ICS1, XM_009106500.3) of SA biosynthesis and the regulatory genes of SA signaling were up-regulated, but the final response genes, such as PR genes, were down-regulated. This phenomenon occurred in the opposite way in the SS inoculation, which supports that the JA and SA hormone pathways are involved in the regulation of JA-SA antagonism depending on the infection of the two *P. brassicae* isolates (Figure 5). Moreover, ET positively affected disease symptoms against *Fusarium oxysporum* [31], and previous studies determined that hypoxia response positively influences gall formation in *Arabidopsis* roots [45,46]. Interestingly, our result also exhibits that the hypoxia-response-related term is more enriched following SS inoculation than that following HS2 inoculation (Figure 3 and Appendix A). Together, these results indicate that the complex signaling network of SA, JA, and ET hormones could be involved in defense mechanisms in Akimeki against the infection of *P. brassicae* isolates. Further studies are required to determine the hormone levels in Alkimeki during the infection with the two *P. brassicae* isolates.

The development of gall in the roots and vascular tissues inhibits water absorption and transport as well as water supply to the aerial region, which result in leaf wilting during the secondary infection of *P. brassicae* [47]. The symptoms are closely linked to stomatal closure, which is associated with abscisic acid (ABA) accumulation and ABA-responsive genes, including RAB18, RD20, and RD22 [48]. Previous research has shown that ABA concentrations in Chinese cabbage root are significantly elevated during the late stage of *P. brassicae* infection compared to the control [49]. Moreover, recent studies determined that the content of ABA in roots was higher in susceptible lines than in resistant lines during the late stages of the infection [34,35]. In line with similar previous results, our results also show that GO terms including GO:0009737 (response to abscisic acid), GO:0009738 (abscisic acid-activated signaling pathway), and GO:0009414 (response to water deprivation) are enriched following SS inoculation compared to HS2 inoculation, whereas the GO:0006833 (water transport) term is highly enriched following HS2 inoculation (Figure 3), suggesting that ABA and ABA-responsive genes could affect water balance by regulating stomata conductance. Our next endeavor is to investigate the ABA pathway-related gene expression and ABA hormone levels in Alkimeki during the different stages of infection.

During the initial step of pathogen infection to plants, PAMPs are sensed by PRRs playing a critical role in the response to pathogen defense on the plant cell’s surface, which transmit signals and lead to PTI [50,51]. Previous studies have reported that chitin elicitor receptor kinase (CERK), brassinosteroid insensitive 1-associated kinase 1 (BAK1), and flagellin sensing 2 (FLS2) trigger PTI [40]. Recent transcriptome approaches revealed that genes involved in PAMPs, Ca^2+^ influx, respiratory burst oxidase homologs (RBOHs), transcription factors, and pathogenesis-related proteins (PRs) are up-regulated in the resistant line against *P. brassicae* in *B. rapa* [25,30]. In contrast to previous results, our findings show that genes involved in PRRs (CERK1, BAK1, EF-Tu mitochondrial, cyclic nucleotide-gated (CNG) ion channel, etc.), PTI (calcium signaling and calcium-binding proteins, calcium-dependent protein kinases (CDPKs), mitogen-activated protein kinase (MAPK) signaling pathway, WRKYs, and PRs, etc.), and ETI (PTI1-like tyrosine protein kinases and receptor-like cytoplasmic kinase 1, etc.) were up-regulated in SS inoculation, whereas only a few of the genes were up-regulated following HS2 inoculation (Figure 4). In addition, pathogen defense-related R genes play an important role in triggering downstream signaling in response to a pathogen penetration [52]. RPS2 (a leucine-rich repeat class), RPS4 (Toll/interleukin-1 receptor-nucleotide-binding site-leucine-rich repeat class), and RPM1 (coiled-coil domain-a central nucleotide-binding site-leucine-rich repeat class) function as R genes in *Arabidopsis thaliana*, which play an immune receptor for defense activation [45,53,54]. PBS1, which is belongs to the protein kinase subfamily, is also an important R gene [46]. Previous studies have determined that RPS2 and PBS1 are up-regulated in the clubroot-resistant variety in *B. rapa* [25]. Our results show that PBS1, RPS2, and RPM1 are more up-regulated following SS inoculation than those following HS2 inoculation (Figure 4), indicating that the R genes may participate in pathogenic response against two *P. brassicae* isolates. The PTI and ETI mechanisms in Akimeki against the SS and HS2 infection may be different from those in the previous studies, which mainly increase in CR plants. The difference between our results and previous studies may result from a different infection stage of pathogen inoculation, the variety of materials, or different effectors of the SS strain from the HS2 strain. Overall, our data suggest that the PTI- and ETI-mediated phosphorylation signaling pathway, Ca^2+^ signaling, ROS production, and WRKY transcription factors play a critical role in the immune response of Akimeki during the phase between primary and secondary infection against the two *P. brassicae* isolates. Our next interest is to understand the intricate defense mechanism underlying the PTI-ETI continuum in Alkimeki during the infection with two *P. brassicae* isolates and at he several different stages of the infection. 

## 4. Materials and Methods

### 4.1. Plant Growth and P. brassicae Field Isolates

Cabbage F_1_ varieties and inbreds used for clubroot resistance to *P. brassica*. The *P. brassicae* Seosan field isolate (SS) and Hoengseong2 field isolate (HS2) were obtained from the naturally occurring club gallbladder used in a previous research paper and used for pathogenicity analysis [7].

### 4.2. Plant Inoculation

Five milliliters of spore suspensions (1.0 × $10^{7}$ spores/mL) of each were prepared as described above, and then they were inoculated via drenching onto the Chinese cabbage cultivars and inbred lines that were grown in 6 × 6 × 6 cm plastic pots (one seedling per pot) for 10 days. The plants were harvested 72 h after inoculation for RNA-seq and qRT-PCR.

### 4.3. RNA Isolation and Library Construction 

The roots of inoculated Akimeki plants were sampled at 0 h (control and Mock) and 72 h after inoculation with the SS isolate and HS2 isolate. Total RNA was isolated using a ${EZ}^{TM}$ Total RNA miniprep kit (Enzynomics, Inc., Carlsbad, CA, USA, Daejeon, Republic of Korea) according to the manufacturer’s protocols. Sequencing libraries were prepared using the Illumina TruSeq RNA Library Prep Kit (San Diego, CA, USA). The purity of each RNA sample was assessed using a Thermo Scientific NanoDrop (Waltham, MA, USA) and Agilent Bioanalyzer (Santa Clara, CA, USA). The library preparation, quality assessment, and sequencing were performed using the Illumina HiSeq 2000 platform from Seeders Co. (Daejeon, Republic of Korea).

### 4.4. RNA Sequencing and Analysis

Raw sequence reads were cleaned using PRINSEQ-lite (v0.20.4) [55]. Cleaned paired-end reads were aligned to the reference *B. rapa* transcript (CAAS_Brap_v3.01) using Bowtie2 [56,57,58]. The read counts and TMM-normalized TPM (trimmed mean of M value-normalized transcripts per million) values were obtained using the RSEM method in the Trinity (v2.12.0) packages [58]. These gene counts for each replication and treatment were calculated to the negative binomial dispersion for differential gene expression analysis using EdgeR (v3.16.5) [59]. Quality checks included the percentage of mapped reads and PCA plots to ensure the consistency of the RNA-seq experiments (Appendix A and Appendix A). Genes were determined to be significantly differentially expressed if they showed a >4-fold change in expression, with a false discovery rate (FDR)-adjusted *p* < 0.001. Heatmap analysis was utilized to visualize and assess the clustering of the data using programs of PtR of the Trinity package and TBtools v2.112 [60].

### 4.5. Functional Annotation Analysis of DEGs

Functional annotation of differentially expressed genes (DEGs) was carried out using the BLASTX program v2.12.0+ (e-value cutoff 1 × 10^−5^) against the *Arabidopsis thaliana* protein database (TAIR10), on https://www.arabidopsis.org, accessed on 3 June 2024. DAVID (2021 Update) was used for Gene Ontology (GO) term enrichment analysis and Kyoto Encyclopedia of Genes and Genomes (KEGG) pathway enrichment analysis [61]. Enriched GO genes were further analyzed using GSEA (version 4.1.0, Broad Institute of Massachusetts Institute of Technology and Harvard, Cambridge, MA, USA) [62].

### 4.6. Correlation Analysis of Quantitative PCR and NGS Data

qRT-PCR was reacted with ${TOPreal}^{TM}$ qPCR 2X PreMIX–SYBR Green with low ROX (Enzynomics, Inc., Carlsbad, CA, USA, Daejeon, Republic of Korea) according to the manufacturer’s protocols. Primers were designed using Primer3 (v. 0.4.0) [63]. Primers used to amplify and quantify specific cDNAs are indicated in Appendix A. Transcript levels of target genes were normalized relative to the mean CT value of the Actin-2(ACT1) and RNA-seq data in the control condition using the 2^−ΔΔCT^ method.

## 5. Conclusions

We investigated the global gene expression profiling of Akimeki’s early immune response to two *P. brassicae* strains, HS2 and SS inoculations, via the transcriptome approach. GO analysis showed that defense response- and defense hormone-related genes are up-regulated following SS inoculation, and KEGG analysis revealed that the gene expression involved in the plant–pathogen interaction pathway of *B. rapa* is mainly up-regulated following SS inoculation, but down-regulated in the HS2 inoculation. This study contributes to the understanding of the intricate mechanisms underlying plant immunity against different *P. brassicae* strains, laying the foundation for future research into targeted breeding strategies and the development of more robust clubroot-resistant cultivars.

## Figures and Tables

**Figure 1 plants-13-02167-f001:**
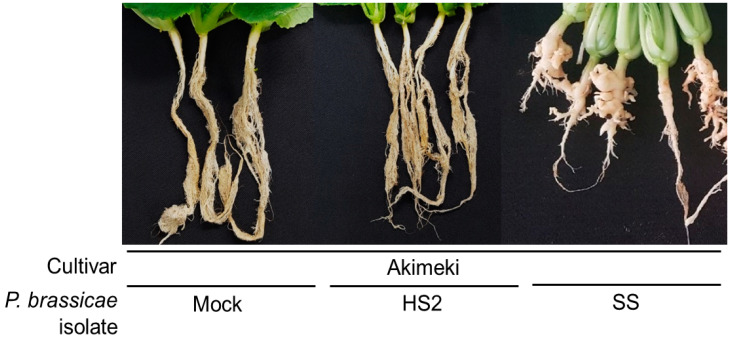
Phenotypes of the inoculation of CR *B. rapa* (Akimeki) with 2 types of *P. brassicae* isolates (Hoengseong2, pathotype 1; Seosan, pathotype 4). The photo was taken five weeks after inoculation.

**Figure 2 plants-13-02167-f002:**
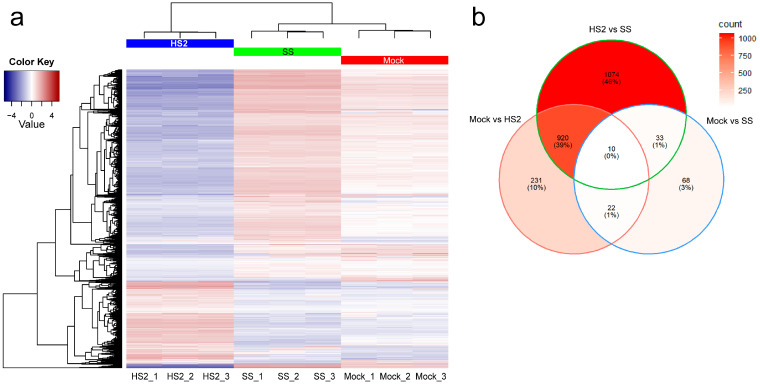
(**a**) Heatmap of relative expression of 2358 DEGs with the indicated treatments. Expression values are log2-transformed median-centered TMM-normalized TPM. Color key indicates Z-scores of expression values. The *x*-axis dendrogram indicates sample similarity and *y*-axis dendrogram indicates the hierarchical clustering of genes with similar expression profiles. Distance and clustering algorithms used for the dendrogram are the complete linkage with Euclidean distances. (**b**) DEG analysis of differentially expressed genes (DEGs) between HS2 inoculation vs. SS inoculation, SS inoculation vs. Mock, and HS2 inoculation vs. Mock.

**Figure 3 plants-13-02167-f003:**
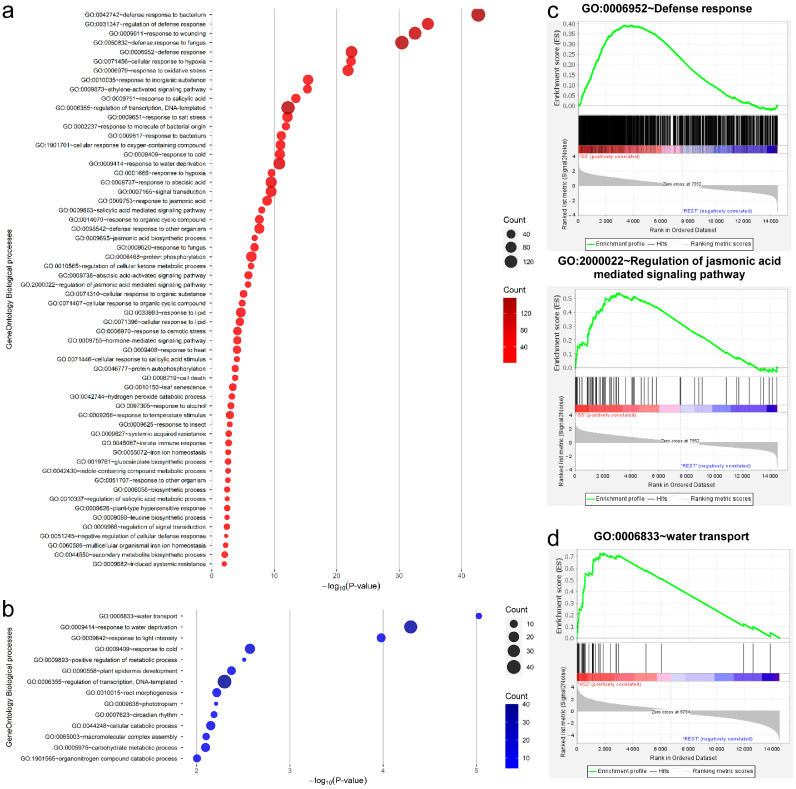
Gene ontology (GO) analysis and gene set enrichment analysis (GSEA) of genes in Akimeki by SS and HS2 inoculation. (**a**,**b**) Enriched GO biological process category of 1524 up-regulated DEGs following SS inoculation and 513 up-regulated DEGs following HS2 inoculation. The size and color depth of the circles represent the number of DEGs. (**c**) Enrichment plot for genes related to the defense responses (GO:0006952~defense response, GO:2000022~regulation of jasmonic acid-mediated signaling pathway). (**d**) Enrichment plot for genes related the water transport (GO:0006833~water transport). GSEA rank was calculated by HS2 vs. REST (Mock and SS). The green line indicates the enrichment profile.

**Figure 4 plants-13-02167-f004:**
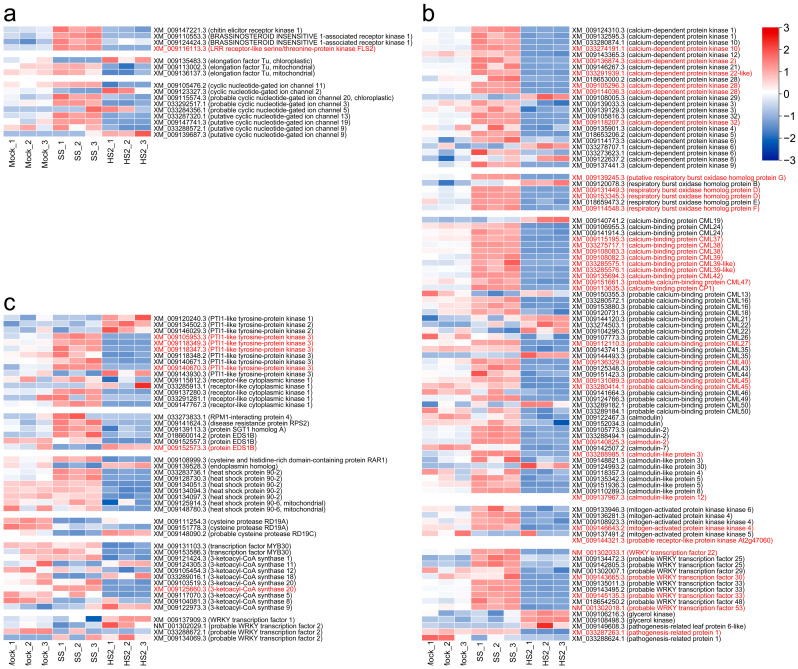
Gene expression heatmaps of genes involved in the plant–pathogen interaction pathway of *B. rapa* in KEGG (Kyoto Encyclopedia of Genes and Genomes). The pathways are organized into three categories: (**a**) the perception of pathogens by pattern-recognition receptors (PRRs), (**b**) pattern-triggered immunity (PTI), and (**c**) effector-triggered immunity (ETI). Heatmap colors represent row-scaled Z-score of TMM-normalized TPM values. The functional descriptions corresponding to the gene are derived from the NCBI Eukaryotic Genome Annotation Pipeline (Annotation release ID: 103.20201202). The gene labels in red indicate DEGs with a fold-change of 4 or more induced by SS and HS2 strains.

**Figure 5 plants-13-02167-f005:**
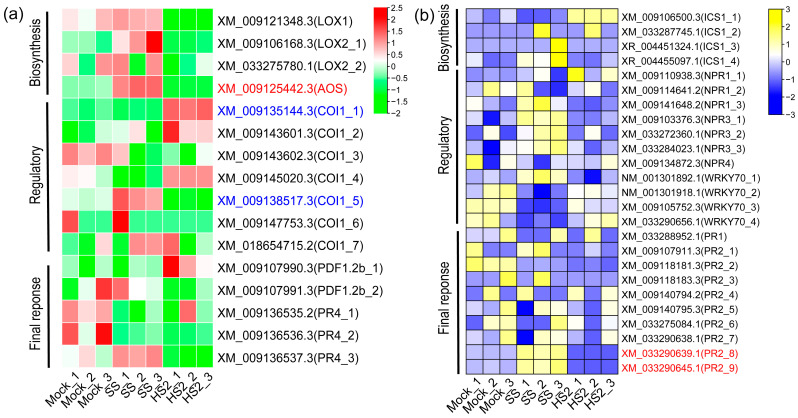
Gene expression patterns related to jasmonic acid and salicylic acid defense pathways in *B. rapa* (Akimeki) against two *P.brassicae* isolates (SS and HS2). Heatmaps of the relative expression of key genes involved in (**a**) JA and (**b**) SA hormones. The colors represent the relatively highly and lowly regulated expression of genes, as indicated by the row-scaled Z-scores of TMM-normalized TPM values. Gene symbols were assigned based on their similarity to Arabidopsis genes, using a BLAST search. Differential gene expression induced by SS and HS2 strains is shown, with genes exhibiting a fold-change of 4 or more labeled in red and those with a fold-change of 2 or more labeled in blue.

**Figure 6 plants-13-02167-f006:**
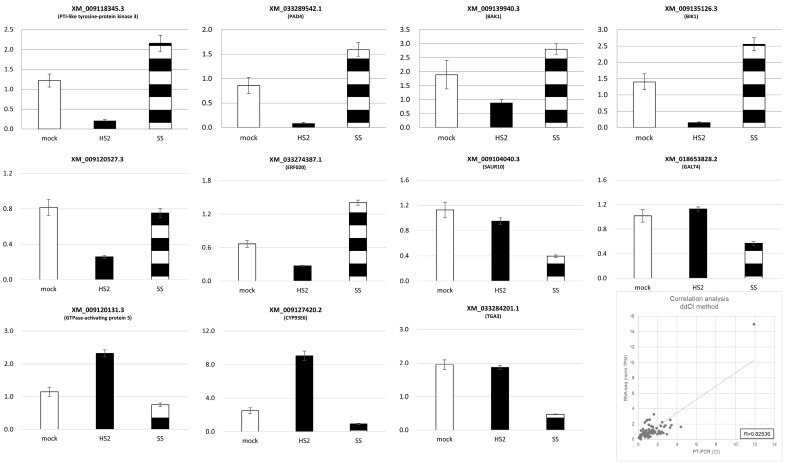
Quantitative real-time PCR validation of the selected genes. Akimeki was inoculated 10 days after sowing, and samples were collected 72 h after inoculation. Isolated RNA was subjected to cDNA synthesis and qRT-PCR. HS2 (Hoengseong2); SS (Seosan). Gene expression values are normalized to BrACT1. Values are the mean and ± standard error of the mean.

## Data Availability

Data is contained within the article or Appendix A. The RNA-seq datasets generated and analyzed during the current study have been deposited in the Sequence Read Archive (SRA) at NCBI under the accession number PRJNA1144072.

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
