# Peer review of "Comparison of Root Transcriptomes against Clubroot Disease Pathogens in a Resistant Chinese Cabbage Cultivar (Brassica rapa cv. ‘Akimeki’)"

_plants, 2024, doi:10.3390/plants13152167_

Round 1

Reviewer 1 Report (Previous Reviewer 1)

Comments and Suggestions for Authors

The authors have significantly improved the manuscript especially with the inclusion of discussion on ABA. There still a few minor issues that need addressing.

1)        PTI stands for pattern-triggered immunity (not PAMP-triggered). Also, there is a recent push in the field to replace the term “PAMPs” with more accurate “MAMPs”, microbe-associated molecular patterns.

2)        Rather than using final response, it would be more accurate to use “output genes” when referring to genes like PR1.

3)        L67 the epitope used for flagellin is flg22 not flg33.

4)        L74-75, PCD and HR refer to the same phenomenon, better to mention only one or state that PCD is also known as HR.

5)        L178-180, can be more specific, COI1 is the JA coreceptor.

6)        Can the authors also check for VSP2 expression, as the output for the MYC (ET-independent) branch of JA pathway? (this is not strictly necessary).

Author Response

Reviewer 1

1)  PTI stands for pattern-triggered immunity (not PAMP-triggered). Also, there is a recent push in the field to replace the term “PAMPs” with more accurate “MAMPs”, microbe-associated molecular patterns.

>> We changed ‘PAMP-triggered immunity’ to ‘pattern-triggered immunity’

2)  Rather than using final response, it would be more accurate to use “output genes” when referring to genes like PR1.

>> We revised Figure 5 using “output genes”.

3)  L67 the epitope used for flagellin is flg22 not flg33.

>> We revised flg22 instead of flg33.

4)  L74-75, PCD and HR refer to the same phenomenon, better to mention only one or state that PCD is also known as HR.

>> We have revised the sentence to: "the ETI associated with programmed cell death (PCD), also known as the hypersensitive response (HR)."

5)  L178-180, can be more specific, COI1 is the JA coreceptor.

>> We have revised the sentence from "whereas COI1, a major regulator of the JA pathway, was mostly upregulated." to "whereas COI1, a major JA co-receptor of the JA pathway, was mostly upregulated."

6)   Can the authors also check for VSP2 expression, as the output for the MYC (ET-independent) branch of JA pathway? (this is not strictly necessary).

>> The B. rapa transcript XM_009131753.3, which is a homolog of the A. thaliana VSP2 (AT5G24770) protein sequence based on BLASTX search, was not expressed in our transcriptome data and therefore was not shown.

Reviewer 2 Report (Previous Reviewer 2)

Comments and Suggestions for Authors

The authors have satisfactorily addressed all my questions.

Author Response

The authors have satisfactorily addressed all my questions.

Reviewer 3 Report (Previous Reviewer 3)

Comments and Suggestions for Authors

Paper by Oh et al reported results of the transcriptome analysis in resistant Chinese cabbage cultivar during development of the Clubroot disease. Authors analyzed disease-resistant cultivar Akimeki reported to be susceptible for some Korean P. brassicae strains. The aim of the work is to study fine-tuned plant signaling by fungal pathogens. Author analyzed three plant transcriptomes obtained at the early stages of infection by Hoengseong2 (HS2, non-pathogenic) and Seosan (SS, pathogenic) P. brassicae isolates and mock.

Major

1. In results section, authors provide data only for DEGs between HS2 and SS (2037 genes). What about mock data? It is not described in results at all. In the Discussion section authors also mention HS2 and SS comparison only (see lines  251-252).

It looks reasonable to involve mock data for analysis results; otherwise, its sequencing seems useless (at least for me, report of this library remained unclear, because it is not used in obtaining of the main results/conclusions of the paper).

For example, it looks reasonable for me to analyze DEGs in HS2 vs mock but not SS vs mock. I suppose that they may reflect some differences between HS2 and SS action at the stage before infection development/immune response (the difference in disease propagation is observed from HS2/SS DEGs). Perhaps authors could suggest any other hypothesis related to mock data to check.

2. The Figure 2 is not properly described in the text. Please extend this figure description. See comment above.

3. Lines 156-158: “To investigate the differences between HS2 and SS inoculation conditions in defense responses, we confirmed the expression of genes corresponding to the plant-pathogen interaction pathway of B. rapa (brp04626) among KEGG pathways by heatmap (Figure 4).” Does authors consider genes from 2037 set (HS2/SS DEGS) in this figure or all genes? If the latter is true, how many genes from 2037 dataset is involved in these pathways and how they can be found in Fig 4? If the former is true, why the information about non-DEGs in this figure is important?

4. Lines 361-362: “Functional annotation of differentially expressed genes (DEGs) was carried out using BLAST program against the Arabidopsis thaliana protein database.” As far as I understood, authors map their sequences to Arabidopsis genes by BLAST, then used Arabidopsis hits in the DAVID to perform GSEA. If so, BLAST parameters should be provided (program BLASTP/BLASTN, method megablast/ blastp-fast or other, e-value / minimal overlap thresholds for hit match). What Arabidopsis sequences were used (genomic database version – TAIR 10 – provide literature ref)?

Did all B. rapa sequences from 2037 DEGs returned Arabidopsis matches? Please provide table of the correspondence between B. rapa and Arabidopsis IDs in the supplement. This is an important point of analysis.

Did the KEGG pathway analysis was performed via such matching? If so, the link between B. rapa and KEGG protein IDs should be provided in the supplement additionally.

 Minor:

5. Line 97: “response of CR to pathogenic and non-pathogenic pathogens” please change to “response of CR to pathogenic and non-pathogenic strains”

6. Figure 2 panel a: does small text above the hierarchical plot (samples vs features) informative? I think it can be removed. Other captions in this panel are hard to read. I recommend to increase the size of this plot (the b panel could be reduced because it contains few details)

7. Some text in Korean appeared when mouse pointer is over Fig 2, 4, 6 in my PDF reader (Foxit). Is it some error or some useful information was not translated?

8. Lines 197-198: “Gene symbols with numbers in parentheses indicate paralogs in B. rapa.”. The sentence is not clear. Please refer to the database from whish these gene symbols were taken. The term paralogs is also unclear. Paralogs are genes in the same genome with different functions (duplicated). Did you mean orthologs (genes from different organism with the same function)? How these gene symbols were determined – using BLAST search or from B. rapa database annotation?

9. Line 251-252: “In this study, we performed the functional analysis on 2,038 DEGs in SS vs. HS2 inoculation out of total DEGs.” In the results section the number of HS2/SS DEGs is 2037.

Comments on the Quality of English Language

I found some typos in the manuscript, please re-read it carefully

Author Response

  1. In results section, authors provide data only for DEGs between HS2 and SS (2037 genes). What about mock data? It is not described in results at all. In the Discussion section authors also mention HS2 and SS comparison only (see lines  251-252).

It looks reasonable to involve mock data for analysis results; otherwise, its sequencing seems useless (at least for me, report of this library remained unclear, because it is not used in obtaining of the main results/conclusions of the paper).

For example, it looks reasonable for me to analyze DEGs in HS2 vs mock but not SS vs mock. I suppose that they may reflect some differences between HS2 and SS action at the stage before infection development/immune response (the difference in disease propagation is observed from HS2/SS DEGs). Perhaps authors could suggest any other hypothesis related to mock data to check.

>> We have performed GO and KEGG enrichment analyses for the DEGs between HS2 inoculation and Mock treatment, specifically focusing on the 231 DEGs that show differences only between these conditions. The results have been included in the Results section and are supported by supplementary table 3.

  1. The Figure 2 is not properly described in the text. Please extend this figure description. See comment above.

>> We have added additional explanations regarding comment 1 in the '2.2 Functional Enrichment Analyses of Differentially Expressed Genes' section.

  1. Lines 156-158: “To investigate the differences between HS2 and SS inoculation conditions in defense responses, we confirmed the expression of genes corresponding to the plant-pathogen interaction pathway of B. rapa (brp04626) among KEGG pathways by heatmap (Figure 4).” Does authors consider genes from 2037 set (HS2/SS DEGS) in this figure or all genes? If the latter is true, how many genes from 2037 dataset is involved in these pathways and how they can be found in Fig 4? If the former is true, why the information about non-DEGs in this figure is important?

>> We selected and visualized defense-related genes from the KEGG pathway (ath04626), including HS2/SS DEGs. Out of the 514 B. rapa genes in the KEGG pathway (ath04626, plant-pathogen interaction pathway), 160 plant defense-related genes were selected and visualized in Figure 4. Among these, 40 genes are part of the 2037 DEGs identified between HS2 and SS inoculation conditions.

  1. Lines 361-362: “Functional annotation of differentially expressed genes (DEGs) was carried out using BLAST program against the Arabidopsis thaliana protein database.” As far as I understood, authors map their sequences to Arabidopsis genes by BLAST, then used Arabidopsis hits in the DAVID to perform GSEA. If so, BLAST parameters should be provided (program BLASTP/BLASTN, method megablast/ blastp-fast or other, e-value / minimal overlap thresholds for hit match). What Arabidopsis sequences were used (genomic database version – TAIR 10 – provide literature ref)?

Did all B. rapa sequences from 2037 DEGs returned Arabidopsis matches? Please provide table of the correspondence between B. rapa and Arabidopsis IDs in the supplement. This is an important point of analysis.

Did the KEGG pathway analysis was performed via such matching? If so, the link between B. rapa and KEGG protein IDs should be provided in the supplement additionally.

>> We identified homologs by conducting a BLASTX search (e-value 1e-5) of B. rapa transcript sequences against the A. thaliana protein sequences from the TAIR10 database. The A. thaliana matches for the HS2/SS DEGs and the KEGG pathway annotations are explained in the Results section and provided in supplementary tables 2 and 3.

 Minor:

  1. Line 97: “response of CR to pathogenic and non-pathogenic pathogens” please change to “response of CR to pathogenic and non-pathogenic strains”

>> We changed the expression of “pathogens” to “strains”

  1. Figure 2 panel a: does small text above the hierarchical plot (samples vs features) informative? I think it can be removed. Other captions in this panel are hard to read. I recommend to increase the size of this plot (the b panel could be reduced because it contains few details)

>> We removed the small text in Figure 2a and adjusted the plot size as mentioned.

  1. Some text in Korean appeared when mouse pointer is over Fig 2, 4, 6 in my PDF reader (Foxit). Is it some error or some useful information was not translated?

>> The content does not include any Korean text. This may be a file format error, so we reattached the original figure file.

  1. Lines 197-198: “Gene symbols with numbers in parentheses indicate paralogs in B. rapa.”. The sentence is not clear. Please refer to the database from whish these gene symbols were taken. The term paralogs is also unclear. Paralogs are genes in the same genome with different functions (duplicated). Did you mean orthologs (genes from different organism with the same function)? How these gene symbols were determined – using BLAST search or from B. rapa database annotation?

>> The gene symbols of orthologs are based on a BLAST search using A. thaliana protein sequences. We have updated the figure legend to include this explanation.

  1. Line 251-252: “In this study, we performed the functional analysis on 2,038 DEGs in SS vs. HS2 inoculation out of total DEGs.” In the results section the number of HS2/SS DEGs is 2037.

>> We revised the number of DEGs in SS vs HS2.

Round 2

Reviewer 3 Report (Previous Reviewer 3)

Comments and Suggestions for Authors

The authors answered all the questions about the previous version of the manuscript. The paper could be accepted for plants. 

Comments on the Quality of English Language

Minor editing of English language required

Author Response

The manuscript requires some minor editing for the English language and tidying up,

l.22 - delete first 'and': ...strains, Seosan (SS)...

>> We deleted ‘and’

l.23 - replace 'in' with 'following SS and HS2 infection'

>> We replaced with 'following SS and HS2 infection'

l.25 - replace 'in' with 'following'

>> We checked whole manuscript and replaced with ‘following’.

Title: Akimeki should not be in italics.

>> We corrected in the title.

Cultivar names usually carry quotes, e.g. 'Akimeki'.

>> We added quotes for 'Akimeki'.

This manuscript is a resubmission of an earlier submission. The following is a list of the peer review reports and author responses from that submission.

Round 1

Reviewer 1 Report

Comments and Suggestions for Authors

The authors have performed transcriptomic study comparing the response of Akimeki variety to its infectious (SS) and non-infectious (HS2) P. Brassicae strains. Through DEG comparison the authors try to identify possible mechanisms of how some P. Brassicae strains are able to infect clubroot resistant varieties. While the study is potentially very valuable and can reveal significant insights, the authors have stopped short of formulating a coherent hypothesis. Furthermore, there seems to be a fundamental confusion about the plant immune response hierarchy in the manuscript. A more detailed and through analysis of the obtained data is needed before the manuscript is suitable for publication.

Specific comments:

The biggest missed opportunity in this manuscript is the more in-depth analysis of the crosstalk between JA and SA signaling pathways during the infection. P. Brassicae is an obligate biotroph, defense against which would be mediated by SA. The SA and JA signaling tend to favor reduced and oxidized cellular environments, respectively (despite oxidative bursts required for initiation of SA-defense). Unsurprisingly, pathogens often exploit this antagonism. For example, both necrotrophy Botrytis (10.1111/j.1365-313X.2011.04706.x) and herbivores (10.3389/fpls.2013.00113) exploit the SA-JA antagonism (specifically redox) to cause host susceptibility. The increase of JA signaling and RBOHs during the susceptible infection suggests to me that a possible opposite scenario possibly occurs, where biotrophic pathogen is inducing JA signaling to suppress SA signaling and cause virulence. This is already well-known in bacterial infection, where JA-mimicking coronatin is produced by bacteria to prevent stomatal closure and defense by suppressing SA signaling. The authors should really do more literature review and try to formulate a coherent hypothesis. Unfortunately, pretty much the only thing the manuscript is stating now is that immune pathways are involved in this infection process, which is self-evident.

As a starting point, the authors can look at how are key SA and JA synthesis genes regulated in response to SS and HS2 infections? The authors should look at their homologs (ICS and AOS, respectively). This would give much better information about what is happening, rather than looking at downstream genes which have multiple regulatory inputs. The authors already mentioned that SA is known to be accumulated stronger in susceptible plants based on prior research, however, it is worth checking for their own dataset.

Conversely, how are the final outputs of the pathways regulated? PR1,2 and 5 come into mind for SA, and VSP2 and PR4 for JA. The analysis so far is limited to intermediate regulators, such as BIK1 and BAK1 (PTI kinases).

While these results might still discount JA-SA antagonism hypothesis, it still needs to be tested.

Additionally, it is striking that water transport is increased in the resistant strain. Do the authors know why this might be? Perhaps more discussion on this (maybe ABA signaling is involved?) is needed.

Lines 66-73: The zig-zag immunity model presented here is incorrect. Indeed, the first layer of defense is PAMP-triggered immunity (PTI), which is a relatively limited response. However, PAMPs are not necessarily secreted or part of cell wall. The most prominent bacterial PAMPs are flagellin (flg22) and elongation factor (elf18). To subvert PTI, pathogens can secrete effectors (not PAMPs!) into the cell, through, for example, type 3 secretion system in bacteria, causing effector-triggered susceptibility (ETS). Sometimes effectors can be recognized by intracellular NB-LRR receptors (R genes) evolved in response to ETS, to trigger a stronger, ETI response often accompanied by programmed cell death. There is also an issue of toxins, which can be neither PAMPs nor effectors, the most obvious example of which would be bacterial coronatin, which mimics JA.

Line 165: Similarly, while there has been recent evidence of mutual potentiation of PTI and ETI, PAD4 is most strongly (and directly) associated with TIR-domain NB-LRR mediated ETI, and not PTI.

Line 225: BAK1 is a kinase, not PRR receptor

Overall, while I don’t want to discourage the authors and dismiss their hard work, I think the manuscript, as it is now, is too rushed and misses in-depth analysis to be considered complete. If more analysis is performed, then the manuscript has a potential of bringing forth a significant advance in our understanding of clubroot disease.

Comments on the Quality of English Language

While the English is generally passable, there are still several things to improve, for example:

L38: no need for “a” for cell elongation

L46: again, no need for “a” (it’s never used with plural)

L47: no need to repeat “by clubroot disease”

L67: should be PAMPs (plural)

L134 (and elsewhere): It’s GO terms, not GO genes. Alternatively, “Genes belonging to XXX GO term category”

passable, there are still several things to improve, for example:

L38: no need for “a” for cell elongation

L46: again, no need for “a” (it’s never used with plural)

Reviewer 2 Report

Comments and Suggestions for Authors

Oh et al. presented Comparison of root transcriptomes against Clubroot-disease 2

pathogens in resistant Chinese cabbage cultivar (Brassica rapa 3 cv. Akimeki)

Line 91:The chitin-signaling pathway induces the expression of 78 early chitin-responsive genes such as PKs, WRKYs, ERFs, and ZFs through the MAPK 79 cascade, and regulates the expression of defense genes such as PRs, RLKs, and NB-LRRs - spell them out

Line 102: Figure 1- it would be interesting to show the above-ground tissue too.

Line 133: Spell out GSEA. What are the differences between GO analysis and GSEA analysis?

Figure 3 c-d is a little bit hard to read the details of the plots, and get understand what the trend means. Authors may consider improving both the visibility and the descriptions of these figures.

Author Response

Please the attachment.

Reviewer 3 Report

Comments and Suggestions for Authors

Paper by Oh et al describes the results of RNA-seq analysis of the root transcriptomes against Clubroot-disease pathogens in resistant Chinese cabbage cultivar (Brassica rapa cv. Akimeki).

However, this research has serious bioinformatics flaws (see Section 4.4. RNA sequencing and analysis).

First and the foremost,  authors did not use splice-aware alignment tool for mapping their reads. They used Bowtie2 which is aimed at genome reads alignment only. It cannot correctly align transcriptome reads to the reference genome. See for example review https://link.springer.com/article/10.1186/s13059-016-0881-8 (Box 3. Mapping to a reference).

Second, authors did not provide information about quality check of their alignments. No percentage of mapped reads, no PCA plot of the samples/replicates to demonstrate the consistency of the RNA-seq experiments.  

Thus, all the results obtained in the article are not credible.

Therefore this paper cannot be accepted to Plants. 

Reviewer 4 Report

Comments and Suggestions for Authors

Dear Author,

Thank you for your great work. Your study addresses a significant agricultural problem in resistant cultivars—clubroot disease. The use of RNA sequencing to explore early molecular responses is a common approach that could yield valuable insights for breeding resistant cultivars.

The abstract is generally clear but could benefit from slight reorganization for readers.

Initial mention of CR varieties could be confusing at first present, specifying that Akimeki is a CR variety in front would help if I understood correctly.

Lack of sequencing data presentation, a brief mention of the total reads, mapping ration and correlation in the method or results would strengthen the manuscript comprehensiveness.

Overall, the manuscript offers valuable insights into the molecular mechanisms of clubroot resistance in Chinese cabbage. The findings have potential applications in breeding. With minor revisions for clarity and organization, the manuscript will interest readers and comprehensiveness.

Comments on the Quality of English Language

Minor editing of English language required
